# Prospectors: Leveraging Short Contexts to Mine Salient Objects in High-dimensional Imagery

**Gautam Machiraju** [1]  **Arjun Desai** [2]  **James Zou** [1]  **Christopher Ré** [3]  **Parag Mallick** [4]

## Abstract

High-dimensional imagery consists of high-resolution information required for end-user decision-making. Due to computational constraints, current methods for image-level classification are designed to train with image *chunks* or down-sampled images rather than with the full high-resolution context. While these methods achieve impressive classification performance, they often lack visual grounding and, thus, the *post hoc* capability to identify class-specific, salient objects under weak supervision. In this work, we (1) propose a formalized evaluation framework to assess visual grounding in high-dimensional image applications. To present a challenging benchmark, we leverage a real-world segmentation dataset for *post hoc* mask evaluation. We use this framework to characterize visual grounding of various baseline methods across multiple encoder classes, exploring multiple supervision regimes and architectures (*e.g.*, ResNet, ViT). Finally, we (2) present *prospector heads*: a novel class of adaptation architectures designed to improve visual grounding. Prospectors leverage chunk heterogeneity to identify salient objects over long ranges and can interface with any image encoder. We find that prospectors outperform baselines by upwards of +6 balanced accuracy points and +30 precision points in a gigapixel pathology setting. Through this experimentation, we also show how prospectors can enable many classes of encoders to identify salient objects without re-training and also demonstrate their improved performance against classical explanation techniques (*e.g.*, Attention maps).

[1]Department of Biomedical Data Science, Stanford University [2]Department of Electrical Engineering, Stanford University [3]Department of Computer Science, Stanford University [4]Department of Radiology, Stanford University. Correspondence to: Gautam Machiraju <gmachi@stanford.edu>.

*Workshop on Interpretable ML in Healthcare at International Conference on Machine Learning (ICML)*, Honolulu, Hawaii, USA. 2023. Copyright 2023 by the author(s).

## 1  Introduction

High-dimensional imagery (*e.g.*, megapixel and gigapixel) is common in multiple domains and necessitates new capabilities from the models we develop. The challenge of dimensionality is particularly evident in domains spanning pathology (Gurcan et al., 2009), autonomous driving (Yurtsever et al., 2020), remote sensing (Coffey, 2012), and cosmology (Pesenson et al., 2010). In these domains, models are frequently trained and validated for image-level classification. However, domain expert decisions often rely on important higher-order tasks, which are implicitly defined but seldom explicitly validated. One such task is the identification of salient, class-specific objects, referred to as ***weakly supervised salient object detection*** (SOD) (Choe et al., 2020; Wang et al., 2021; Borji et al., 2019).

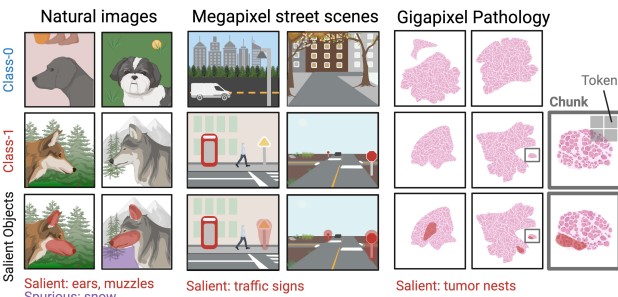

**Figure 1:** Salient objects in various vision domains. Ideally, vision models can identify these objects *in situ* without explicit training.

SOD examines a model's capability to make ***visually grounded*** predictions, *i.e.*, those that (A) accurately localize salient objects in an image despite training on only image-level labels and (B) attribute quantitative class-specific scores to said objects. Visually grounded models are of general importance to better interpret predictions, but are of particular need in settings where:

i. data distributions between classes overlap greatly or where class differences only manifest as small salient objects (*i.e.*, multiple instance assumption (Amores, 2013; Carbonneau et al., 2016));

ii. predictions are used in tandem with regions of interest (created either by domain experts or model explanations) to make decisions (*e.g.*, medical diagnoses);

iii. annotations are cost-prohibitive and only image-level labels are generated;

iv. or salient objects are not fully known to domain experts and necessitate concept discovery (*e.g.*, biomarkers in multiplexed pathology images (Echle et al., 2020; Lewis et al., 2021)).

Current models for image-level classification are largely incapable of processing images at full-context and struggle to reliably identify salient objects. Current convolutional neural networks (CNNs) (Simonyan & Zisserman, 2014; He et al., 2015; Liu et al., 2022) struggle to process high-dimensional inputs due to modern GPU memory constraints while Vision Transformers (ViTs) (Dosovitskiy et al., 2020) additionally experience quadratic time and space complexity in sequence length (Keles et al., 2022). As a workaround, these models are typically trained to classify images in a hierarchical manner: input images are first commonly broken down into *chunks* (*i.e.*, patches) (Nazeri et al., 2018; Chen et al., 2022; Campanella et al., 2019; Jean et al., 2019), which are then respectively processed with an encoder's receptive fields or as constituent tokens (Figure 1). After constructing chunk-level representations, those representations are used for image-level classification. SOD is then typically performed with *post hoc* model explanations and evaluated qualitatively. Despite achieving state of the art classification performances, no model to our knowledge is quantitatively evaluated for SOD capabilities in real-world, high-dimensional settings. In synthetic settings (Machiraju et al., 2022), chunk-based models without image-level context have been shown to perform SOD with low precision, where all or no chunks are predicted as salient.

To address the need for visually grounded models in high-dimensional imagery, we make two main contributions. First, we propose a prescriptive **evaluation framework** that re-purposes real-world segmentation datasets for SOD and tests a model's visual grounding. We additionally use this framework to evaluate baseline SOD approaches, including Attention Maps. To expand evaluation from that of previous studies (Machiraju et al., 2022), we propose new metrics that holistically summarize SOD performance over various binarization thresholds (a central post-processing step). Secondly, we present *prospector heads*: a novel class of adaptation approaches that leverages chunk heterogeneity learned by upstream encoders (*i.e.*, chunk-trained CNNs or ViTs), finds class-differential motifs over the corpus of image chunks, and localizes salient objects at an image-level. We show that prospectors can be parameter-lite and data-efficient, do not require encoder back-propagation or retraining (*i.e.*, is "plug-in ready" (Kim et al., 2019)), are encoder-agnostic (*i.e.*, agnostic to architecture, embedding dimension, learning regime), and generalize to multiple modalities. Finally, we compare prospectors to current explanation methods to show that the coupling of encoder

inference and prospector adaptation can achieve higher quality SOD and offer increased stability to thresholding.

## 2 Related Work

**Chunk-based modeling for vision:** The community has proposed a variety of CNN and ViT encoders to perform chunk-based modeling for high-dimensional vision applications. Weakly supervised learning (WSL) approaches predict a chunk's class membership typically by using fuzzy labels — *e.g.*, its source image's class label (Machiraju et al., 2022; Halicek et al., 2019; Roy et al., 2019; Hou et al., 2015; Campanella et al., 2019; Nazeri et al., 2018). Despite independent chunk predictions, such models have seen state of the art image classification performance. However, few works have explored WSL-based SOD (Machiraju et al., 2022). On the other hand, unsupervised learning (USL) approaches like *tile2vec* (Jean et al., 2019) have been proposed to encode distributional semantics. While USL methods can encode spatial dependencies, they are currently unable to perform SOD due to their lack of a classification heads. Newer modeling strategies have prioritized learning longer contexts in a hierarchical manner: where upstream encoders feed chunk-level embeddings to downstream encoders for image-level predictions. Examples include WSL-based encoders with downstream transfer learning (Campanella et al., 2019) or self-supervised learning (SSL) with WSL-based fine-tuning of a classifier head (Chen et al., 2022). However, to our knowledge, these approaches have not been tested for their SOD capabilities.

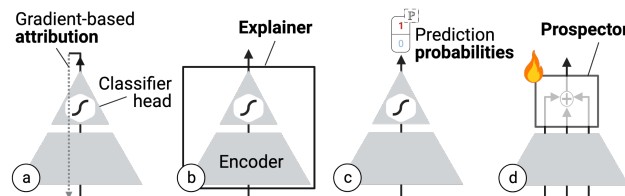

**Figure 2:** Taxonomy of SOD methods, where (a)-(c) are existing methods categories. A flame icon denotes trainable parameters.

**Model explanation & evaluation:** Traditionally, *post hoc* model explanations have been used to detect salient objects for qualitative assessment and model debugging. Explanations usually take the form of input-specific ***explanation maps*** in vision applications, generated using (Figure 2): (a) model-specific, gradient-based attribution methods (*e.g.*, Saliency (Simonyan et al., 2013), Class Activation (Selvaraju et al., 2016), and Attention maps (Jetley et al., 2018)) or (b) model-agnostic Explainers (*e.g.*, SHAP (Lundberg & Lee, 2017), LIME (Ribeiro et al., 2016)) that typically use perturbation-based attribution. However, due to poor visual grounding, the community has recently shifted its language to describe gradient-based attribution methods — away from "explanatory" to "exploratory" (Atrey et al., 2019) or rather as correlational "trends in how predictions are related to

features" (Rudin, 2019). In particular, saliency maps suffer from visual noisiness (*e.g.*, when non-salient objects contain positive pre-activation values (Kim et al., 2019)) and have been qualitatively shown to reflect a classifier's reliance on spurious correlations and shortcuts (DeGrave et al., 2021). Quantitative evaluations of gradient-based attribution have also been conducted using segmentation-like metrics. However, multiple methods have been found to beirreproducible and unrepeatable (Arun et al., 2020), have SOD performance highly correlated with image classification performance (Saporta et al., 2022), and suffer from an inability to accurately localize objects compared to human annotators (Arun et al., 2020; Saporta et al., 2022), especially with small salient objects (Saporta et al., 2022). Furthermore, multiple methods drastically degrade in SOD performance with the removal of image features (Hooker et al., 2019). Attention maps, in particular, have also been suggested as unfaithful to a model's reasoning processes (Jain & Wallace, 2019) and untrustworthy and unreliable via human study experiments (Akula & Zhu, 2022). While there have been techniques to improve SOD localization (*e.g.*, using explanation maps as training guides (Asgari et al., 2022), optimization constraints (Ross et al., 2017), or post-processing techniques (Adebayo et al., 2018)), few works enforce meaningful class-indicative scoring for detected salient objects (Samek et al., 2017).

**Explanations for high-dimensional imagery:** In high-dimensional settings, explanations are typically generated by concatenating the outputs of (a) gradient-based attribution methods applied each image chunk (Campanella et al., 2019; Machiraju et al., 2022; Chen et al., 2022). In addition to gradient-based methods, (c) prediction probabilities for each chunk have been concatenated and used as explanations (Figure 2c) (Campanella et al., 2019; Halicek et al., 2019; Machiraju et al., 2022). However, these approaches do not capture long-range dependencies between image chunks, and can result in low-precision SOD, *i.e.*, with either low recall or low specificity (Machiraju et al., 2022).

## 3 Methods

### 3.1 Salient object desiderata & the SOD task

We first define our desiderata for predicted salient objects:
A. **Localized** — correct (*e.g.*, precise) and contiguous
B. **Class-indicative** — carry scores that indicate class in some way, *e.g.*, ⊕ values for class-1 and ⊖ values for class-0

These desired properties can be assessed with weakly supervised salient object detection (SOD), the task of identifying salient (*i.e.*, class-specific) objects within a datum. SOD is typically implicit to some explicit task (*e.g.*, classification) and can be conceptualized as the *zero-shot capability*

to perform segmentation. For example, in the context of imagery, the SOD task is to detect salient superpixels with image-level classification labels. By withholding ground truth annotations until inference-time, we evaluate a model's grounding capabilities under weak (*i.e.*, coarse) supervision. We refer to a model as *visually grounded* if it can perform the SOD task. While the following presentation of methods is tailored to high-dimensional imagery, it can be generalized to any unstructured data modality.

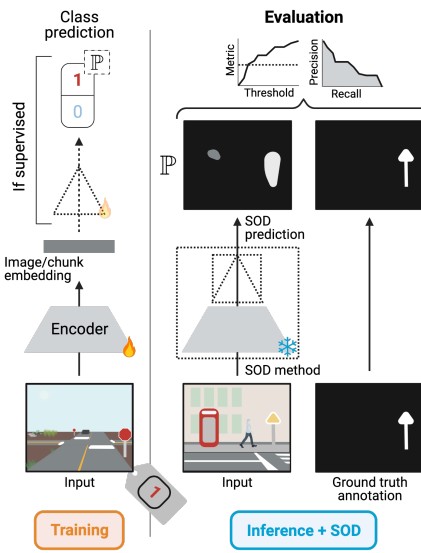

**Figure 3:** SOD pipeline & proposed evaluation. Categories of SOD methods are depicted in Figure 2.

**Task setup & encoder training:** Given some image input $\mathbf{X} \in \mathbb{R}^{H \times W \times D}$, class label $y$, and associated ground truth annotations $\mathbf{Y} \in \{0,1\}^{H \times W}$, a chunk encoder $f_\theta$ is trained to map a data chunk $\mathbf{x} \in \mathbb{R}^{\epsilon \times \epsilon}$ to an embedding $\vec{\mathbf{z}} \in \mathbb{R}^d$. Specifically in the WSL regime, $f_\theta$ is attached to a classifier head $g_\beta : \vec{\mathbf{z}} \mapsto y$, where both are jointly trained to construct a mapping $f_\theta \circ g_\beta : \mathbf{x} \mapsto y$. Ground truth $\mathbf{Y}$ is used only for a *post hoc* evaluation at inference-time.

**Model inference & SOD:** At inference-time, frozen models are used to generate dense prediction maps for salient objects, $\widehat{\mathbf{Y}}$. If using traditional explanation maps for SOD (*e.g.*, prediction probabilities, gradient-based attribution, Explainers), chunk-level explanations must be concatenated to form $\widehat{\mathbf{Y}}$. Due to storage constraints, each chunk's average Saliency or Attention score is used to summarize the chunk's importance as performed by Machiraju et al. (2022). Before evaluation, maps are then post-processed with a variety of thresholding and normalization techniques. We outline these choices in §3.2.1, §3.2.2, and §3.2.3.

### 3.2 SOD evaluation framework

Our proposed evaluation framework is similar to the one described by Machiraju et al. (2022), but is customized for

real-world datasets with ground truth annotations. Instead of centering SOD correctness evaluation around adaptive thresholding (§3.2.1), our framework proposes several forms of evaluation including correctness evaluation over multiple thresholds (§3.2.2) and structural evaluation (§3.2.3). Because the combination of concatenated explanation maps and adaptive thresholding have been associated with potentially low-precision SOD (Machiraju et al., 2022), we believe our additional evaluations can reveal a broader picture of performance.

**Ground truth preparation:** Given the height-width compression performed by encoders, we down-sample ground truth annotations $\mathbf{Y}$ (via inter-area interpolation) to match the $h \times w$ dimensions of the dense predictions $\widehat{\mathbf{Y}}$. This level of granularity makes evaluation computationally feasible.

### 3.2.1 CORRECTNESS EVALUATION VIA ADAPTIVE THRESHOLDING

Adaptive thresholding is a common output-specific technique to binarize explanation maps prior to qualitative or quantitative evaluation (Borji et al., 2019; Machiraju et al., 2022; Adebayo et al., 2018). A popular threshold is double the mean Attention or Saliency score, *i.e.*, $t^* = \frac{2}{n} \sum_i^n \widehat{\mathbf{Y}}_i$, where $n$ is the number of evaluable points. Given a resized ground truth annotation $\mathbf{Y} \in \{0, 1\}^{h \times w}$, we evaluate binarized outputs $\widehat{\mathbf{Y}} > t^*$ and $\widehat{\mathbf{Y}} < t^*$ of class-1 test-set images using some metric $m(\cdot)$ to get an average score $\bar{s}_{t^*}$:

$$\bar{s}_{t^*} = \frac{1}{N} \sum \left[ m(\widehat{\mathbf{Y}} > t^*, \mathbf{Y}) \right],$$

and similarly for the other inequality direction. We use metrics such as balanced accuracy, Matthews correlation coefficient (MCC), and precision given the rarity of salient objects within class-1 images.

### 3.2.2 CORRECTNESS EVALUATION VIA MULTI-THRESHOLDING

While adaptive thresholding is popular, it can be sensitive to outlier values. To test $\widehat{\mathbf{Y}}$'s performance more holistically, we conduct multi-thresholding as an additional inexpensive set of evaluations for SOD. This evaluation approach calculates test-set scores over a small number of thresholds, and then averages them per threshold $\bar{s}_t$. We choose thresholds $t \in [0, 0.1, \ldots, 1]$, yielding a coarse trajectory of performance and thus allowing us to evaluate an output's **stability to thresholding**. For a given encoder-SOD pipeline, we define stability as:

$$\Delta s = \max(\{\bar{s}_t \, \forall t\}) - \min(\{\bar{s}_t \, \forall t\}).$$

We also support the use of near-continuous multi-thresholding (Borji et al., 2019), often for AUROC and

AUPRC computation, as a comprehensive evaluation for SOD correctness. This evaluation is carried out for each feature-scale normalized dense prediction $\widehat{\mathbf{Y}}$ against its corresponding ground truth $\mathbf{Y}$ to yield average scores over the class-1 examples in the test-set. For Attention maps, we perform feature-scale normalization over the absolute value of Attention scores.

### 3.2.3 CONTIGUITY EVALUATION

To help identify failure modes, we also propose a simple metric to quantitatively measure the contiguity and dispersal of predicted salient objects in $\widehat{\mathbf{Y}}$. We define the **mean object dispersal** (MOD) as the average object size divided by the number of detected objects. In practice for the image modality, we detect objects as connected components (with 8-way connectivity) (Bolelli et al., 2020) and use bounding box areas to approximate object size. Our implementation of MOD thus takes on values in $[0, h \times w]$.

## 3.3 A new approach to SOD: prospector heads

To address the need for visual grounding and both (A) localized and (B) class-indicative SOD, we present *prospector heads*: a trainable class of adapters that leverage chunk embeddings and long-range information to predict salient objects with only coarse supervision (Figure 2d). Because high-dimensional datasets typically contain only a few hundred or thousand labeled examples, prospectors in these settings should be parameter-lite to prevent overfitting. To this effect, we present a simple, statistically-driven prospector called K2 — a parameter-lite, data-efficient, computationally efficient, encoder-agnostic, and threshold-stable method for finding salient objects. K2 is modality-generalizable given its use of graphical data structures and is presented as such.

**K2 prospectors:** At a high-level, K2 draws from two main algorithmic inspirations. Firstly, it draws from methods that mine *shapelets*, the discriminative sub-sequences in time-series (Ye & Keogh, 2009; Grabocka et al., 2014; Guillaume et al., 2021). Secondly, K2 draws from the self-Attention mechanism and its ability to attribute importance to inputs as it learns to solve a downstream task (Vaswani et al., 2017). However, to realize parameter-lite modeling, K2 draws on mathematical and statistical machinery to compute chunk importance and attribution — including vector quantization, clustering, graphs, skip-grams, and linear models. **K2 is named after its hashing data structure**, $K_k$, a self-complete graph with $k$ vertices. First, we define some preliminary concepts:

**Definition 3.1** (Map Graph). A map graph $G(V, E)$ is a collection of vertices $V$ and edges $E$ connecting neighboring vertices in real-space. Each vertex $v^{(i)} \in V$ has features $\vec{\mathbf{f}}^{(i)}$ and an edge $e^{(i \leftrightarrow j)} \in E$ connects vertices $v^{(i)}$ to $v^{(j)}$.

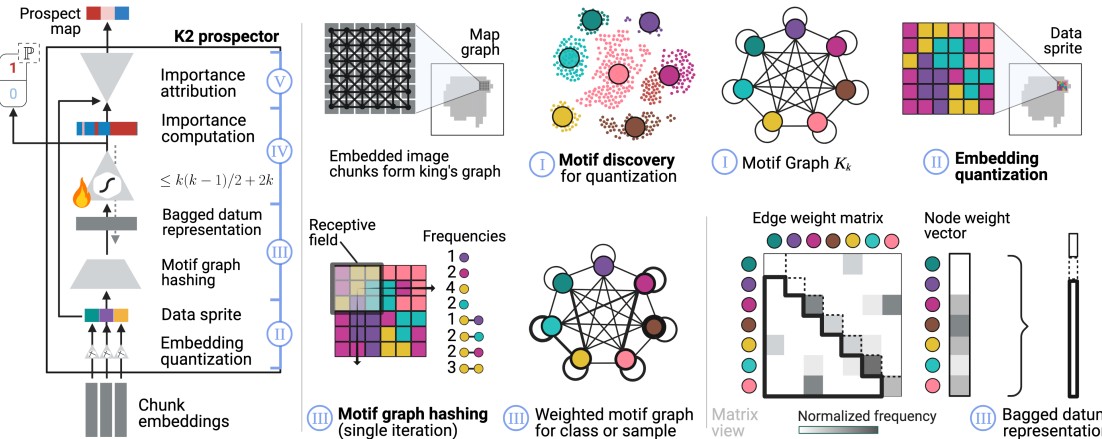

**Figure 4:** K2 prospector circuit diagram with intermediate data structures. A flame icon denotes trainable parameters. Boldface indicates algorithmic step.

**Definition 3.2** (Complete Graph). A complete graph $K_k(V, E)$ is a collection of $k$ vertices $V$ and $k(k-1)/2$ edges $E$. Every pair of distinct vertices $v^{(i)}$ to $v^{(j)}$ are connected by a unique edge $e^{(i\leftrightarrow j)}$, making it fully connected.

**Definition 3.3** (Self-complete Graph). A complete graph $K_k(V, E)$ is also a self-complete graph if it also contains all self-edges that map from any one vertex $v^{(i)}$ to itself with edge $e^{(i\leftrightarrow i)}$. Thus, self-complete graphs contain $k$ vertices and $k + k(k-1)/2$ edges.

**Problem setup:** First, we set up the problem scenario offered by upstream encoders. Suppose we have chunk embeddings $\vec{\mathbf{z}}^{(i)}\ \forall i$ for a datum. We can then represent any datum as a map graph $G$ where each vertex $v^{(i)}$ represents a chunk embedding with features $\vec{\mathbf{f}}^{(i)} := \vec{\mathbf{z}}^{(i)} \in \mathbb{R}^d$. In the context of imagery, our datum is an image $\mathbf{X}$. Each image is broken into chunks, where each chunk $\mathbf{x}^{(i)}$ is encoded as an embedding $\vec{\mathbf{z}}^{(i)}$ by an encoder $f_\theta : \mathbf{x}^{(i)} \mapsto \vec{\mathbf{z}}^{(i)}$. In the imagery setting, $G$ specifically resembles a map graph known as a *king's graph* (depicted in top row of Figure 4) — *i.e.*, where vertex coordinates follow a Cartesian grid and all neighbors are connected.

**I. Motif discovery & motif graph instantiation:** This step first takes a population-level perspective by taking a random sampling of chunk embeddings and then creating a representative embedding space. This embedding space is then partitioned into $k$ sub-spaces using some clustering method $\psi$ (*e.g.*, $k$-means). Each subspace then represents one of $k$ motifs found in the population. K2 then constructs *motif graph* $K_k$ (Figure 4), a self-complete graph that that reflects the $k$ motifs (*i.e.*, sub-spaces) as its nodes. The edges of $K_k$ represent the combinatorial space of motif-motif spatial associations that may occur within the data, as explained further in proceeding steps.

**II. Embedding quantization:** Next, K2 assigns motif IDs to each of $G$'s vertices to reduce dimensionality. Namely, $\psi$

is used to quantize each feature vector $\vec{\mathbf{z}}^{(i)}$ such that $\vec{\mathbf{z}} \mapsto s$, where $s \in \mathbb{S} = \{1, \ldots, k\}$, *i.e.*, $\psi : \mathbb{R}^d \to \mathbb{S}$ (Figure 4). This new map graph $\tilde{G}$ has vertices $v^{(i)}$ has a single feature $s^{(i)}$ for all $i$. For convenience, we refer to $\tilde{G}$ as a ***data sprite*** given its relatively small dimensionality compared to its source gigapixel image (*i.e.*, data compression ratio on the order of $10^9$) and discrete set of $k$ possible values (Figure 4). For intuition, heterogeneity of the sprite is parameterized by the choice of $k$ in Step I.

**III. Motif graph hashing:** Next, K2 performs motif graph hashing — the process of encoding a sprite-level or class-level representation for $\tilde{G}$. At its core, K2 constructs ***weighted motif graphs***, where each node and edge weight is respectively a normalized frequency of a motif or motif-motif pair. This computation is performed by scanning a data sprite $\tilde{G}$'s vertices with a local neighborhood search. Given a window size (*i.e.*, a receptive field) $r$, K2 gathers the frequencies of all motifs and motif-motif pairs (*i.e.*, $r$-skip-2-grams) within a window. Vertices $v^{(i)}$, edges $e^{(i\leftrightarrow j)}$, and self-edges $e^{(i\leftrightarrow i)}$ are thus respectively weighted with the frequencies of the $k$ motifs, the $k(k-1)/2$ motif-motif pairs of distinct motifs, the $k$ motif-motif pairs for a shared motif. A ***bagged datum representation*** $\vec{\mathbf{B}}$ is constructed by concatenating the vertex, self-edge, edge weights (Figure 4). The formalization of this step can be found in Algorithm 1.

**IV. Importance computation:** This next step computes the global importance of each motif or motif-motif pair using the bagged representation $\vec{\mathbf{B}}$. Multiple models can be used — for example, we can train a linear classifier $g_\beta : \vec{\mathbf{B}} \mapsto y$ to learn over the dataset's bagged representations and use its learned coefficients $\beta$ as motif and motif-motif importance. Another example includes computing mean bagged representations per class (*i.e.*, $\vec{\mathbf{B}}_0$ and $\vec{\mathbf{B}}_1$) and instead conducting differential expression analysis (Anders & Huber, 2010), *i.e.*, a hypothesis test for independent means and significant fold-changes $\beta$. Either variant is

parameter-lite and thus data-efficient: the former only contains $2k + k(k-1)/2$ learnable parameters, while the latter is parameter-free.

**V. Importance attribution for prospect maps:** After computing importance $\beta$, we spatially attribute them using the same sliding skip-gram search used in Step III. We first define a ***prospect map***, a map graph $\dot{G}$ with the same topology as $\widetilde{G}$ and scalar features initialized at zero for all vertices. For a given neighborhood of size $r$ centered on vertex $v^{(i)}$, we add the constituent importance scores for each occurrence of a motif or motif-motif pair. In this manner, we build in class-specific meaning via the $\oplus$ or $\ominus$ signage of summed scores. For intuition, $r$ parameterizes the level of *smoothing* over the K2 map by modulating the number of chunks used for attribution. In the context of imagery, $\dot{G}$'s vertex coordinates map directly to form the elements of an array $\widehat{\mathbf{Y}}$. A formalization is provided in Algorithm 2.

**Computational efficiency:** Because K2 only relies on forward passes from $f_\theta$, our approach to SOD is 3-4× more computationally efficient than gradient-based attribution methods. Our analysis is based on empirical results from computing Floating Point Operations (FLOPs) for model inference (*i.e.*, using forward passes only). Since K2's FLOPs can be considered negligible, this efficiency boost is approximated by the forward-backward pass FLOP ratio of 1:2 to 1:3 (Zhou et al., 2021; Baydin et al., 2017). This efficiency is especially relevant for multi-billion parameter Foundation Models (Bommasani et al., 2021) that could be used as upstream encoders.

# 4 Experiments & Results

**Dataset:** We center this work on Camelyon16, a benchmark dataset of 400 gigapixel pathology images (270 train, 130 test) of breast cancer metastases in sentinel lymph nodes (Ehteshami Bejnordi et al., 2017). The goal of this dataset is to identify the metastases as salient superpixels. All images were chunked (size $\epsilon = 224$) and filtered for foreground tissue regions (as opposed to the glass background). This process resulted in more than 200K unique chunks without augmentation. Dataset characteristics are described further in §A.2 and Figure A.1.

**Encoders:** To demonstrate K2's utility, we paired it with multiple encoders described in Table 1. We train two encoders from scratch (USL-trained tile2vec (Jean et al., 2019) and a WSL-trained ViT) and also use generalist and domain-specific vision-language models operating on image chunks: CLIP (Radford et al., 2021) and a pathology fine-tuned version of CLIP called PLIP (Huang et al., 2023). More details can be found in §A.3. To provide SOD baselines when possible, we generate ***stitched probability maps*** (SPMs) and ***stitched Attention maps*** (SAMs) — *i.e.*, the concatenation

of chunk prediction probabilities or mean attention scores (Figure 2b,c). SPMs for CLIP and PLIP were generated with the queried text-encoded labels: "normal lymph node" for class-0 and "lymph node metastasis" for class-1.

| Encoder Alias | Architecture | Learning Regime | Training Epochs | Embed Size ($d$) |
|---|---|---|---|---|
| tile2vec | ResNet-18 | USL | 20 | 128 |
| ViT | ViT/16 | WSL | 30 | 1024 |
| CLIP | ViT-B/32 | SSL | ✗ | 512 |
| PLIP | ViT-B/32 | SSL | ✗ | 512 |

**Table 1:** Experimented encoders. Pre-training denoted by ✗.

## 4.1 K2 grid search & multi-thresholding results

To test K2's SOD performance over multiple configurations, we conducted an extensive grid search over hyperparameters (*e.g.*, $k, r$) to train and evaluate 325 prospectors per encoder (*i.e.*, 1300 in total). Further details are outlined in §A.3. For each of these prospectors, we use multi-thresholding (§3.2.2) to conduct evaluation with greater granularity than adaptive thresholding but without the computational overhead of near-continuous thresholding. Results are summarized in Table 2 and reflect the best metric performance over sampled thresholds, along with stability (§3.2.2) in parentheses. These results are also depicted pictorially in §A.5. Reported numbers in the bottom half reflect the best performances over all 325 K2 configurations per encoder.

Our results first verify the need for correctness metrics that are imbalance-aware. Standard metrics are relatively uninformative because of the small salient objects in our test-set. For example, accuracy is generally high due the rarity of salient chunks. Secondly, we confirm that current SOD baselines (top half of Table 2) have relatively high recall and low precision when compared to K2 — where the latter sees gains as large as +30 points in precision and +6 points in balanced accuracy. The preference toward precision in the precision-recall trade-off is further contextualized by the MCC metric. While K2's detected objects achieve some element-by-element correlation with ground truth, baseline MCC scores reflect zero correlation. This lack of correlation, coupled with high recall, indicates that baselines have a tendency to indiscriminately flag objects as salient (as seen in the top right of Figure 6). Thirdly, we observe K2's superior threshold-stability over numerous encoders and metrics — often seeing only a fraction of the change in scores over the sampled thresholds.

## 4.2 Image classification vs SOD

To further probe K2's constructed representations, we assess the relationship between image classification and SOD. K2 prospectors classify images based on their underlying model for importance computation (Step IV in §3.3) — as suggested, we implemented both linear model and differential expression prospector variants. For simplicity, differen-

| Encoder -SOD | Standard Metrics | | | | Metrics for Imbalance | |
|---|---|---|---|---|---|---|
| | Accuracy | Dice | Precision | Recall | Balanced Accuracy | MCC |
| ViT-SAM | 0.829 (0.703) | 0.179 (0.179) | 0.144 (0.144) | **0.691** (0.691) | 0.5 (0.162) | 0.0 (0.413) |
| ViT-SPM | 0.829 (0.704) | **0.191** (0.191) | 0.161 (0.161) | **0.691** (0.691) | 0.5 (0.154) | 0.0 (0.408) |
| CLIP-SPM | 0.829 (0.699) | 0.179 (0.179) | 0.144 (0.144) | **0.690** (0.690) | 0.5 (0.152) | 0.0 (0.343) |
| PLIP-SPM | 0.829 (0.699) | 0.179 (0.179) | 0.152 (0.152) | **0.690** (0.690) | 0.5 (0.152) | 0.0 (0.343) |
| **tile2vec-K2** | 0.840 (0.185) | 0.167 (0.078) | 0.226 (0.102) | 0.448 (0.187) | 0.504 (0.048) | 0.014 (0.058) |
| **ViT-K2** | 0.840 (0.195) | 0.161 (0.069) | 0.213 (0.082) | 0.450 (0.188) | 0.510 (0.052) | 0.023 (0.070) |
| **CLIP-K2** | 0.863 (0.093) | 0.196 (0.111) | 0.431 (0.217) | 0.410 (0.157) | 0.543 (0.033) | 0.115 (0.062) |
| **PLIP-K2** | 0.879 (0.097) | 0.196 (0.122) | 0.408 (0.256) | 0.431 (0.162) | 0.559 (0.036) | 0.134 (0.073) |

**Table 2:** Results of multi-thresholding correctness evaluation (§3.2.2). Parentheses contain score stability $\Delta s$ for each metric. Top half are baseline methods. Key: Dark green indicates a top score, light green indicates a second best score. Higher is better for all metrics.

tial expression variants apply two decision rules directly on feature-scale normalized K2 maps: (i) the maximum value after Gaussian smoothing and (ii) the mean of the top-five chunk values, both followed by thresholding at 0.5. Linear model variants apply these same two decision rules and also run inference using their trained parameters. For each K2 configuration, we report maximum achieved classification AUROC and AUPRC and plot these metrics alongside SOD precision and balanced accuracy in §A.7.

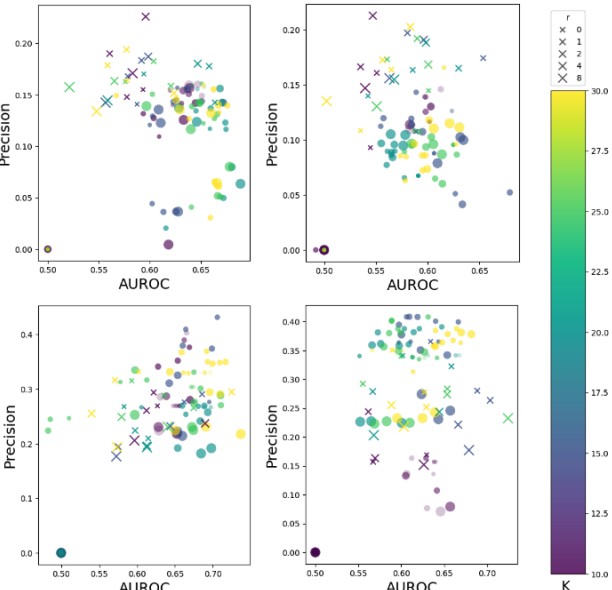

**Figure 5:** Scatter plots of 325 prospector configurations and their performance on image classification (AUROC) vs SOD (precision). Left-to-right then top-to-bottom: tile2vec, ViT, CLIP, PLIP. Importance computation is denoted by marker type: differential expression as ○, linear model as ×. Hyperparameter $k$ is denoted by marker color and $r$ is reflected by marker size.

Figure 5 hones in on classification AUROC and SOD precision and reveals how differently all experimented encoders behave. First, looking at SOD precision alone (y-axis), we observe that parameter-free differential expression prospector variants (marker style ○) are consistently the most precise among CLIP and PLIP encoders, while linear model vari-

ants (marker style ×) are more precise for tile2vec and ViT encoders. This suggests that the former two encoders can perform long-range tasks *nearly for free*, as they require less sophisticated embedding adaptation to localize salient objects. Secondly, looking at both axes together, we observe various relationships between classification AUROC and SOD precision among the encoders. For tile2vec and ViT encoders, we qualitatively observe a trade-off between axes, suggesting a Pareto Frontier. This behavior may suggest that these two encoder-SOD pipelines are not precluded from relying on spurious correlations or shortcuts as depicted in Figure 6 and §A.6. In contrast, the vision-language encoders *do not experience* as strong of a trade-off between axes and even suggest some correlation for the CLIP encoder — supporting more advanced reasoning processes.

Turning to hyperparameters, we also see divergent impacts based on encoder choice. CLIP and PLIP generally experience highest SOD precision with small window sizes $r$ (denoted by marker size), while classification AUROC experiences less of a direct relationship with window size. This same phenomenon does not hold for tile2vec and ViT encoders, as larger window sizes achieve the highest SOD precision in both cases, perhaps implying lower heterogeneity in their data sprites (Figure 6) and a need for more context. Overall, the top-right corner of each encoder's scatter plot suggests that small $r$ and smaller $k$ do achieve the optimal choices for both SOD precision and classification AUROC.

### 4.3 Prospector selection for additional evaluation

Using the grid search results, we selected top-performing K2 prospectors per metric in Table 2 to perform both adaptive thresholding (§3.2.1) and near-continuous thresholding (§3.2.2) for correctness evaluation, as well as structural evaluation (§3.2.3). Example K2 and baseline maps are displayed in Figure 6 and in §A.6, reiterating how K2 is able to localize tumor metastases to high degrees of correctness, while baselines either suffer from extremely low precision (*e.g.*, ViT-SAM) or even seem to systematically make reversed predictions in certain test-set images (*e.g.*,

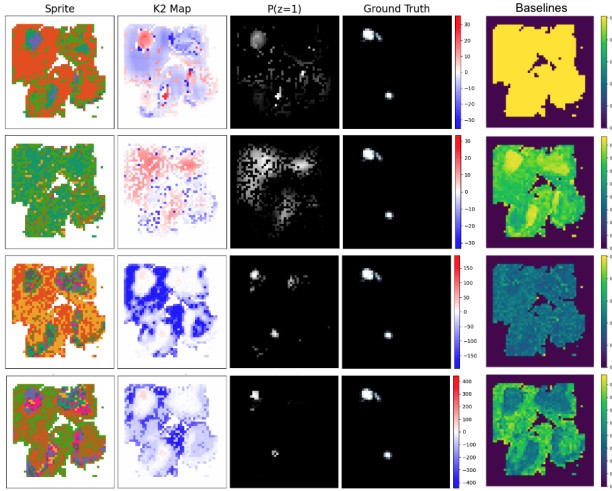

**Figure 6:** K2 maps from highest precision configurations via multi-thresholding. In row-descending order for first four columns: tile2vec, ViT, CLIP, PLIP. Under the baseline column (right), we show unprocessed ViT-SAM, ViT-SPM, CLIP-SPM, PLIP-SPM.

PLIP-SPM). Looking at Table 3, we first observe that adaptive thresholding (left-hand side) allows for improved precision for baselines, but decreased precision, balanced accuracy, and MCC for K2 when comparing against the multi-thresholding results in Table 2. This may suggest that the drastically different construction of K2 prospect maps are not as amenable to standard adaptive thresholding. Despite this dip in performance, K2 still achieves superior SOD correctness in all three metrics compared to baselines. Looking at near-continuous thresholding, K2 achieves dominant AUROCs and AUPRCs, where even the tile2vec encoder with K2 is able to outperform SPMs generated by CLIP and PLIP. Finally, in structural evaluation, K2 demonstrates an ability to construct contiguous (*i.e.*, low-dispersion) SOD predictions, where baselines all achieve MOD scores of zero — thus, implying the identification of many non-contiguous salient chunks. In particular, tile2vec is able to construct salient object predictions with significantly more contiguity, likely due to the spatial awareness offered by its loss function. Because only a small subset of 24 prospectors were used in this stage of analysis, these results can be considered conservative estimates of optimal performance.

## 5   Discussion & Conclusion

In this work, we present a SOD evaluation framework and prospector heads, a new class of methods that can perform SOD in high-dimensional imagery with state of the art performance. Our proposed prospector, K2, is parameter-lite, data-efficient, and modality-generalizable. Despite its simplicity, K2 outperforms baselines over a suite of evaluations and advances encoders' abilities to predict salient objects in a (A) localized (*i.e.*, precise and contiguous) and (B) class-indicative manner. Furthermore, we show that K2 is

threshold-stable and encoder-agnostic with consistent gains in SOD performance over baselines. This simple class of prospectors even provides the SOD capability to unsupervised models. We contextualize our findings below:

**Foundation Models and prospectors can identify salience in high-dimensional data.**   Both the general-purpose and domain-specific vision-language models (CLIP and PLIP, respectively) very clearly benefit from boosted performance with K2 — potentially due to their rich, heterogeneous embedding spaces. In addition, we observed that the vision-language models experienced less stark of a trade-off between image classification and SOD performance when compared to tile2vec and ViT (Figure 5, §A.7). These findings support the utility of short-context, prospector-enabled Foundation Models for long-range tasks like SOD in high-dimensional data. Furthermore, the computational efficiency of prospectors will enable SOD at scale.

**Where do prospectors and SOD fit into explainable AI?** We show that prospectors can perform SOD and that SOD helps us formalize the task of aligning the user's and model's concepts of salience. However, prospectors cannot be considered model explanations in the strictest sense due to their adaptation of intermediate embeddings — and thus, prospect maps are *unfaithful* (Ross et al., 2017; Jacovi & Goldberg, 2020) to the upstream encoder's reasoning process. K2's hybridized approach should be noted, however: it makes use of global explanations to make local, input-specific (Adadi & Berrada, 2018; Das & Rad, 2020; Ras et al., 2020) SOD predictions. Given these clarifications, we *can* conclude that K2 provides *plausible* (Jacovi & Goldberg, 2020) salient objects and potential *interpretability* (Ross et al., 2017) by relying on the semantic understanding of an encoder's embedding space. Hopefully this potential for *informativeness* (Lipton, 2016) via plausible SOD and human interpretation inspire new forms of prospector heads.

**Healthcare implications of this work**   are centered around fostering model interpretability and the capability for models to *instruct users* to identify salient regions within biomedical data despite coarse supervision. As seen in this work, one specific application is in accelerating computer-assisted diagnostics with improved interpretability behind whole slide classification in gigapixel pathology. In addition, this work can be used in settings without ground truth (*e.g.*, multiplexed imagery) to enable the discovery of novel prognostic biomarkers (Echle et al., 2020; Lewis et al., 2021). Finally, our method's modality-generalizable formulation will also spur generalized implementation for future modalities, wherever chunk-based encoders are common: large-scale text (Reimers & Gurevych, 2019), graphs (Derry & Altman, 2022), and time-series (Ju et al., 2021; Wang & Xu, 2022). These modalities open up many possible ap-

| | Adaptive | | | Near-continuous | | | Structural |
|---|---|---|---|---|---|---|---|
| Encoder -SOD | Precision | Balanced Accuracy | MCC | AUROC | AUPRC | AP | MOD |
| ViT-SAM | $0.171^*$ | 0.5 | 0.0 | 0.348 | 0.152 | 0.161 | 0.0 |
| ViT-SPM | $0.171^*$ | 0.5 | 0.0 | 0.396 | 0.185 | 0.195 | 0.0 |
| CLIP-SPM | $0.171^*$ | 0.5 | 0.0 | 0.296 | 0.146 | 0.156 | 0.0 |
| PLIP-SPM | $0.171^*$ | 0.5 | 0.0 | 0.293 | 0.149 | 0.158 | 0.0 |
| **tile2vec-K2** | $0.167^*$ | 0.501 | 0.004 | 0.439 | 0.255 | 0.183 | **4.061** |
| **ViT-K2** | $0.167^*$ | $0.502^*$ | $0.005^*$ | 0.490 | 0.274 | 0.190 | 0.103 |
| **CLIP-K2** | 0.244 | 0.523 | 0.075 | 0.544 | 0.333 | 0.243 | 1.669 |
| **PLIP-K2** | 0.251 | 0.522 | 0.077 | 0.559 | 0.481 | 0.238 | 1.788 |

**Table 3:** Results of adaptive thresholding correctness (§3.2.1), near-continuous thresholding correctness (§3.2.2), and structural evaluation. An asterisk (*) indicates a higher score achieved by backward thresholding. Key: Dark green indicates a top score, light green indicates a second best score. Higher is better for all metrics.

plications: identifying salient sentences in large corpora of biomedical literature, binding pockets in protein structures, or even clinically relevant anomalies in electroencephalography signals, respectively.

**Future work** includes expanding K2's functionality, understanding K2's behavior, implementing additional baselines, and applying K2 to other image datasets and even other modalities. Regarding K2's functionality, we plan to explore 3-mer hashing (*i.e.*, $r$-skip-3-grams). To understand K2 behavior, we will mathematically characterize different encoders' embedding spaces (Figure A.2) to identify correlations with performance. Further, we will study the relationship between image classification and SOD and add to the debate on the relationship between predictive performance and interpretability (Rudin, 2019; Rudin et al., 2021). Regarding vision baselines, we plan to include additional text queries for vision-language SPMs and also plan to compare K2 with parameter-heavy prospectors (*e.g.*, Attention-based). Finally, we plan to shift our current imagery-centered implementation to a modality-generalizable implementation to enable a wide range of applications.

**Limitations** of this work include those around hyperparameter selection. Future work will study effect on chunk size $\epsilon$ over all encoders, as SOD and image classification performance will likely exhibit interesting behavior depending on this choice. Future work will also employ a validation set for hyperparameter tuning to allow for more definitive test-set comparisons.

**Code access:** Our code is publicly available at `https://github.com/gmachiraju/lofi`.

**Acknowledgements:** We thank Elliot Epstein, Michael Poli, Benjamin Spector, Armin Thomas, and Sarah Hooper for helpful discussions and feedback. We additionally thank the Stanford Data Science Scholarship program and the Canary-ACED Graduate Fellowship for trainee support.

We gratefully acknowledge support under 5R01CA249899, 1R01AG078755, as well as the support of NIH under No. U54EB020405 (Mobilize), NSF under Nos. CCF1763315 (Beyond Sparsity), CCF1563078 (Volume to Velocity), and 1937301 (RTML); US DEVCOM ARL under No. W911NF-21-2-0251 (Interactive Human-AI Teaming); ONR under No. N000141712266 (Unifying Weak Supervision); ONR N00014-20-1-2480: Understanding and Applying Non-Euclidean Geometry in Machine Learning; N000142012275 (NEPTUNE); NXP, Xilinx, LETI-CEA, Intel, IBM, Microsoft, NEC, Toshiba, TSMC, ARM, Hitachi, BASF, Accenture, Ericsson, Qualcomm, Analog Devices, Google Cloud, Salesforce, Total, the HAI-GCP Cloud Credits for Research program, the Stanford Data Science Initiative (SDSI), and members of the Stanford DAWN project: Facebook, Google, and VMWare. The U.S. Government is authorized to reproduce and distribute reprints for Governmental purposes notwithstanding any copyright notation thereon. Any opinions, findings, and conclusions or recommendations expressed in this material are those of the authors and do not necessarily reflect the views, policies, or endorsements, either expressed or implied, of NIH, ONR, or the U.S. Government.

Figures 1, 2, 3, and 4 were created with `BioRender.com`.

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

# A  Supplementary Material

## A.1  Algorithms behind K2

The pseudo-code here reflects implementations specifically for image data, as reflected in the codebase. Namely, data sprites (described as map graphs $\tilde{G}$) are instead represented as arrays $\mathbf{S}$.

---

**Algorithm 1** Motif Graph Hashing (MGH)

---

**Require:** Sprite $\mathbf{S} \in \{0, 1, \ldots, k\}^{h \times w}$
**Require:** window size $r$
1: Per-category count vector $\vec{\mathbf{F}}$ where
   $\vec{\mathbf{F}}_c = \sum_{i=1}^{H} \sum_{j=1}^{W} \mathbb{1}[\mathbf{S}_{ij} = c]\ \ \forall c \in \{1, \ldots, k\}$
2: Initialize co-occurrence count matrix: $\mathbf{L} \leftarrow \mathbf{0}^{k \times k}$
3: **for** $1 \leq i \leq h$ **do**
4:   **for** $1 \leq j \leq w$ **do**
5:     **if** $\mathbf{S}_{ij} == 0$ **then**
6:       continue
7:     **end if**
8:     $i_s \leftarrow \max(1, i - r), i_e \leftarrow \min(h, i + r)$
9:     $j_s \leftarrow \max(1, j - r), j_e \leftarrow \min(h, j + r)$
10:     **for** $1 \leq c' \leq K$ **do**
11:       $\mathbf{L}_{\mathbf{S}_{ij}c'} \leftarrow \mathbf{L}_{\mathbf{S}_{ij}c'} +$
             $\sum_{i'=i_s}^{i_e} \sum_{j'=j_s}^{j_e} \mathbb{1}[c' = \mathbf{S}_{i'j'}]$
12:     **end for**
13:     $\mathbf{L}_{\mathbf{S}_{ij}\mathbf{S}_{ij}} \leftarrow \mathbf{L}_{\mathbf{S}_{ij}\mathbf{S}_{ij}} - 1$
14:   **end for**
15: **end for**
16: Initialize order-agnostic co-occurrence vector:
   $\vec{\mathbf{L}} = [\mathbf{L}_{ij} + \mathbf{L}_{ji}\ \ \forall i \in 1, \ldots, k, j \in i + 1, \ldots, k]$
17: Concatenate the per-class vector and co-occurrences:
   $\vec{\mathbf{B}} = \vec{\mathbf{F}} \oplus \mathbf{diag}(\mathbf{L}) \oplus \vec{\mathbf{L}}$
18: Normalize the counts: $\vec{\mathbf{B}} \leftarrow \vec{\mathbf{B}} / \sum_i^{|\vec{\mathbf{B}}_i|} \vec{\mathbf{B}}_i$
19: **return** $\vec{\mathbf{B}}$

---

**Algorithm 2** Importance Attribution (IA)

---

**Require:** Sprite $\mathbf{S} \in \{0, 1, \ldots, k\}^{h \times w}$
**Require:** Feature Importance for categories $\beta^F \in \mathbb{R}_+^k$
**Require:** Co-occurrences $\beta^L \in \mathbb{R}_+^{k \times k}$
1: Initialize mask $\mathbf{Y} \leftarrow \mathbf{0}^{h \times w}$
2: **for** $1 \leq i \leq h$ **do**
3:   **for** $1 \leq j \leq w$ **do**
4:     **if** $\mathbf{S}_{ij} == 0$ **then**
5:       continue {# **background chunk**}
6:     **end if**
7:     $\mathbf{Y}_{ij} \leftarrow \mathbf{Y}_{ij} + \beta^F_{\mathbf{S}_{ij}}$
8:     $i_s \leftarrow \max(1, i - r), i_e \leftarrow \min(h, i + r)$
9:     $j_s \leftarrow \max(1, j - r), j_e \leftarrow \min(h, j + r)$
10:     $\mathbf{Y}_{ij} \leftarrow \mathbf{Y}_{ij} + \sum_{c'=1}^{K} \sum_{i'=i_s}^{i_e} \sum_{j'=j_s}^{j_e} \beta^L_{\mathbf{S}_{ij}\mathbf{S}_{i'j'}} \mathbb{1}[c' = \mathbf{S}_{i'j'}]$
11:     $\mathbf{Y}_{ij} \leftarrow \mathbf{Y}_{ij} - \beta^L_{\mathbf{S}_{ij}\mathbf{S}_{ij}}$
12:   **end for**
13: **end for**
14: **return** $\mathbf{Y}$

---

**Algorithm 3** K2 Training & Inference

---

**Require:** Training set of images $\mathcal{X}_{\text{train}}$
**Require:** Window size $r$, cluster number $k$
**Require:** Test image $\mathbf{X}^*$
**Require:** Encoder $f_\theta$
1: $\mathcal{Z}_{\text{sample}} \leftarrow \{f_\theta(\mathbf{x})$ for randomly sampled $\mathbf{x}$s from $\mathcal{X}_{\text{train}}\}$
2: $\psi \leftarrow$ train-$\psi\,(k, \mathcal{Z}_{\text{train}})$
3: **for** $\mathbf{X}$ in $\mathcal{X}_{\text{train}}$ **do**
4:   $\mathbf{Z} \leftarrow \text{concat}\,(\{f_\theta(\mathbf{x}) \,\forall \mathbf{x} \in \mathbf{X}\})$
5:   $\mathbf{S} \leftarrow \text{concat}\,(\{\psi(\vec{z}) \,\forall \vec{z} \in \mathbf{Z}\})$
6:   $\vec{\mathbf{B}} \leftarrow \text{MGH}(\mathbf{S}, r)$ {# **Alg. 1**}
7:   $\beta \leftarrow \text{train}\, g_\beta(\vec{\mathbf{B}})$
8: **end for**
9: $\mathbf{Z}^* \leftarrow \text{concat}\,(\{f_\theta(\mathbf{x}) \,\forall \mathbf{x} \in \mathbf{X}^*\})$
10: $\mathbf{S}^* \leftarrow \text{concat}\,(\{\psi(\vec{z}) \,\forall \vec{z} \in \mathbf{Z}^*\})$
11: $\vec{\mathbf{B}} \leftarrow \text{MGH}(\mathbf{S}^*, r)$
12: $\mathbf{Y} \leftarrow \text{IA}(\mathbf{S}^*, \beta)$ {# **Alg. 2**}
13: **return** $\mathbf{Y}$

---

## A.2    Dataset statistics

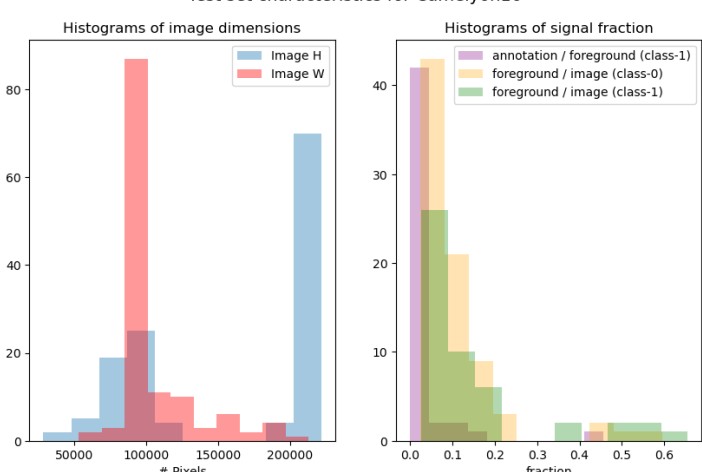

**Figure A.1:** Test dataset statistics.

## A.3    Implementation & experimental details

**Tile2vec encoder training**    Our USL encoder is tile2vec, a ResNet-18 architecture (He et al., 2015) trained for 20 epochs on a single NVIDIA T4 GPU. For training, the training set of 200K chunks were formed into nearly 100K triplets with a sampling scheme similar to that of Jean et al. (2019). These triplets were then used to train tile2vec with the triplet loss function (Jean et al., 2019).

**ViT encoder training**    We trained a custom ViT for trained for 30 epochs on a single NVIDIA T4 GPU. It was trained using independent chunk predictions and fuzzy labeling via image-level labels.

**Embedding quantization**    To quantize embeddings, we use $k$-means clustering over a training dataset-wide random sample ( 4500) of chunk embeddings in order to assign constituent pseudo-labels to data sprites.

**κ2 : motif graph hashing**    We note that the dimensionality of the bagged representation $\vec{\mathbf{B}}$ is in actuality piece-wise with respect to $r$ and $k$:

$$\#\vec{\mathbf{B}} := \begin{cases} k & r = 0 \quad \text{(unigrams)} \\ (2k + k(k-1)/2 & r > 0 \quad \text{(bigrams)} \end{cases}$$

but introduced the bigrams case in the body of the text for simplicity.

After motif graph hashing and formation into bagged representations, we perform TF-IDF normalization (Karen, 1972) on our computed motif and motif-motif frequencies.

**κ2 : importance computation**    Importance computation models $g_\beta$ are implemented as either linear models or differential expression. For linear model variants, we specifically use a LASSO classifier and conduct training over the bagged representations for 3000 iterations.

On the other hand, differential expression variants are parameter-free. These variants instead compute feature importance via class-level differences — specifically by constructing class-level bagged representations (*i.e.*, averaged vectors), computing logarithmic fold-changes (*i.e.*, ratios) between class representations, and testing each feature for significance via a Mann-Whitney U hypothesis test. For filtration after hypothesis testing, we use common thresholds that act as additional hyperparameters: significance threshold $\alpha$ and log2-fold change threshold $\tau$. Prior to filtration, we adjust our chosen significance threshold using the commonly used Bonferroni correction: our original $\alpha$ threshold is divided by the number of motifs and motif-motif pairs, *i.e.*, $\alpha^* = \alpha/\#\vec{\mathbf{B}}$. Finally, to perform filtration, we use $\pm\tau$ to filter out sufficiently small fold changes (*e.g.*, $\pm 1$, which indicates a requirement for doubling in $\log_2$-scale) and use $\alpha^*$ to filter out non-significant differences.

**Grid search details**  To determine the best-performing prospectors, we performed an extensive grid search over all hyperparameters $k$, $r$, $\alpha$, $\tau$ (with the latter two only associated with differential expression). Selected values include $K \in \{10, 15, 20, 25, 30\}$, $r \in \{0, 1, 2, 4, 8\}$, $\alpha \in \{0.01, 0.025, 0.05, 1\}$, and $\tau \in \{0, 1, 2\}$. This search results 25 linear model prospector variants and 300 differential expression variants.

## A.4  Embedding space t-SNE plots

To argue against modeling in a manner without larger image contexts, *i.e.*, with an assumption of independent and identically distributed (IID) chunks, we visualize the embedding spaces of our encoders. The lack of separation of class-1 chunks from class-0 chunks also intuitively depicts the difficulty of using modeling approaches like anomaly detection [c].

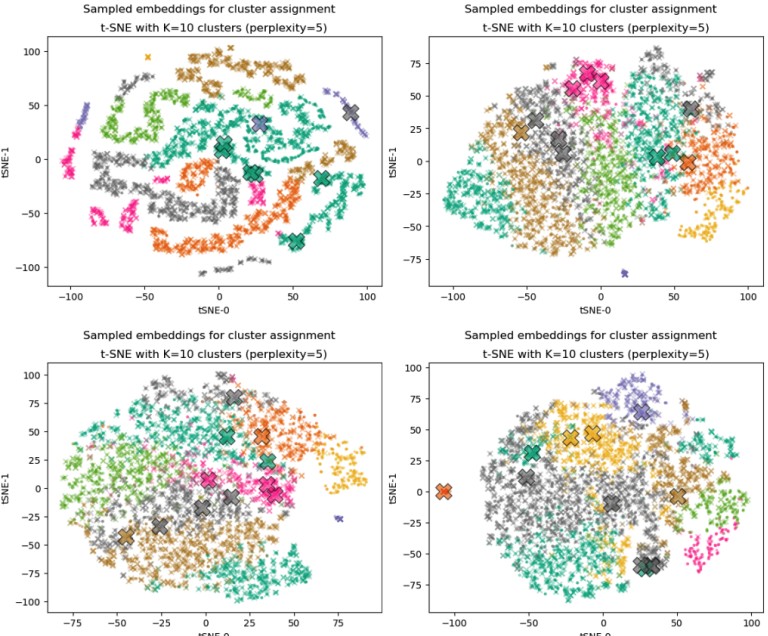

**Figure A.2:** tSNE plots, left-to-right then top-to-bottom: tile2vec, ViT, CLIP, PLIP. Marker color denotes motif label, marker type denotes ground truth annotation for a chunk: ○ for class-0 (non-salient), × for class-1 non-salient, and the much larger ✕ for class-1 salient.

## A.5  Multi-thresholding trajectory plots

Multi-thresholding trajectory plots for each metric. Blue lines track SPM scores and orange lines track SAM scores. Going from left-to-right then top-to-bottom, we display: tile2vec, ViT, CLIP, PLIP.

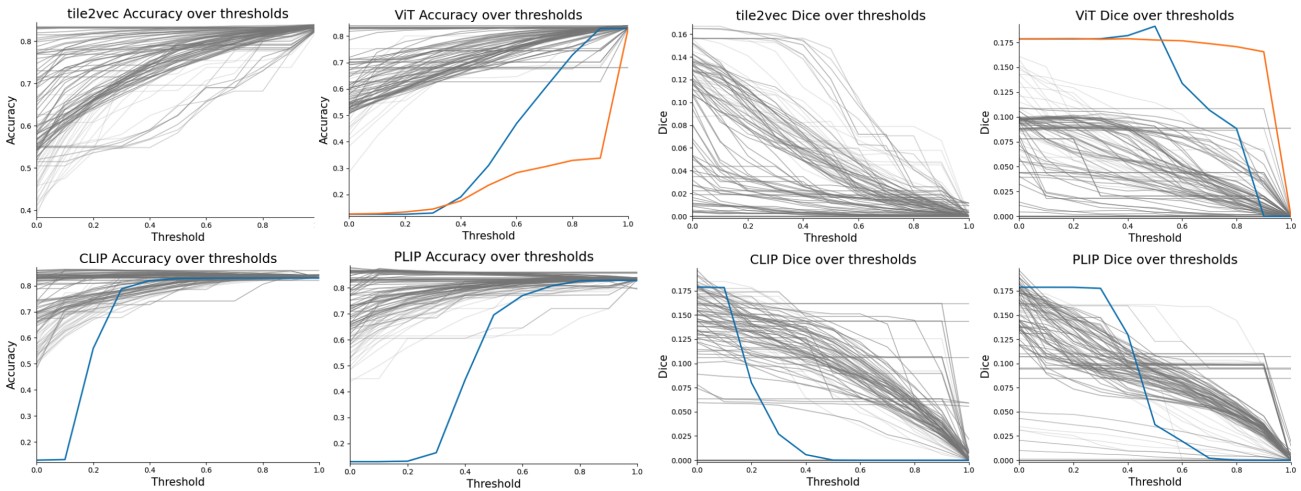

**Figure A.3:** Accuracy.

**Figure A.4:** Dice.

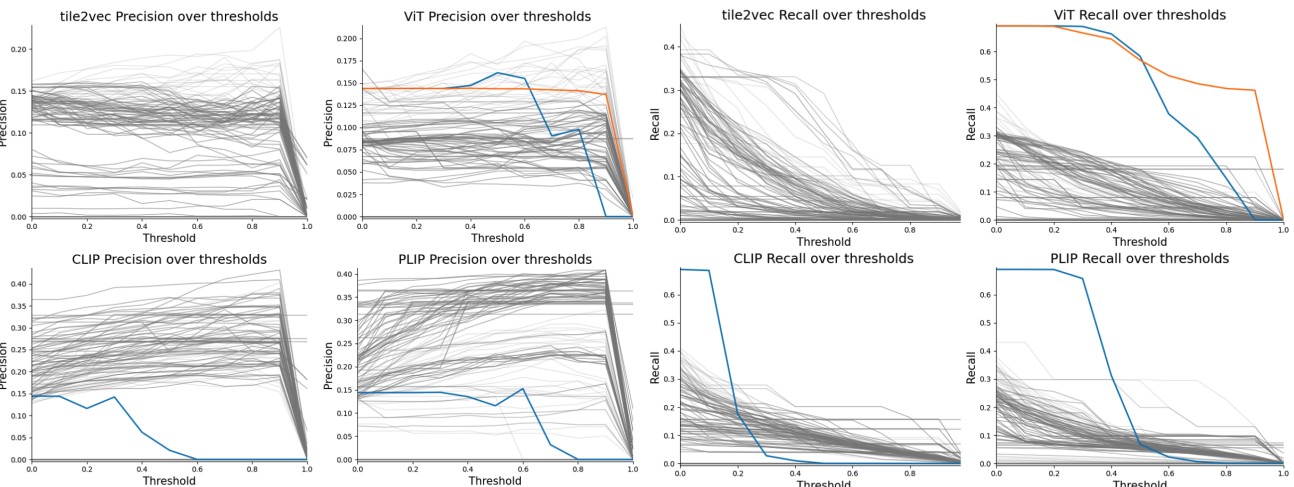

**Figure A.5:** Precision.

**Figure A.6:** Recall.

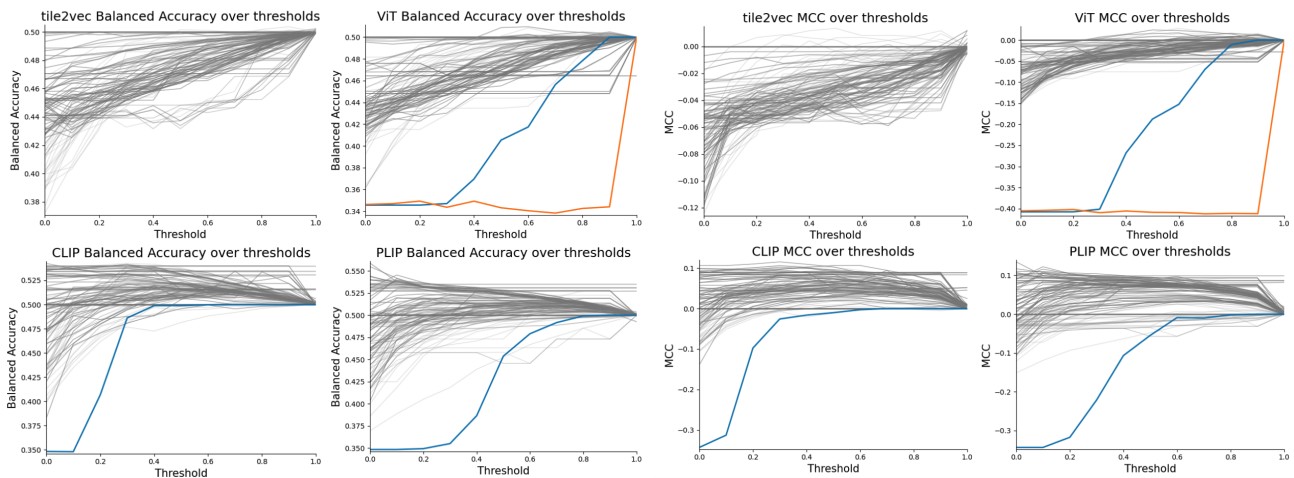

**Figure A.7:** Balanced accuracy.

**Figure A.8:** MCC.

## A.6 Additional ᴋ2 prospect maps

Example K2 maps from the highest scoring configurations determined with multi-thresholding. In row-descending order, we display: tile2vec, ViT, CLIP, PLIP.

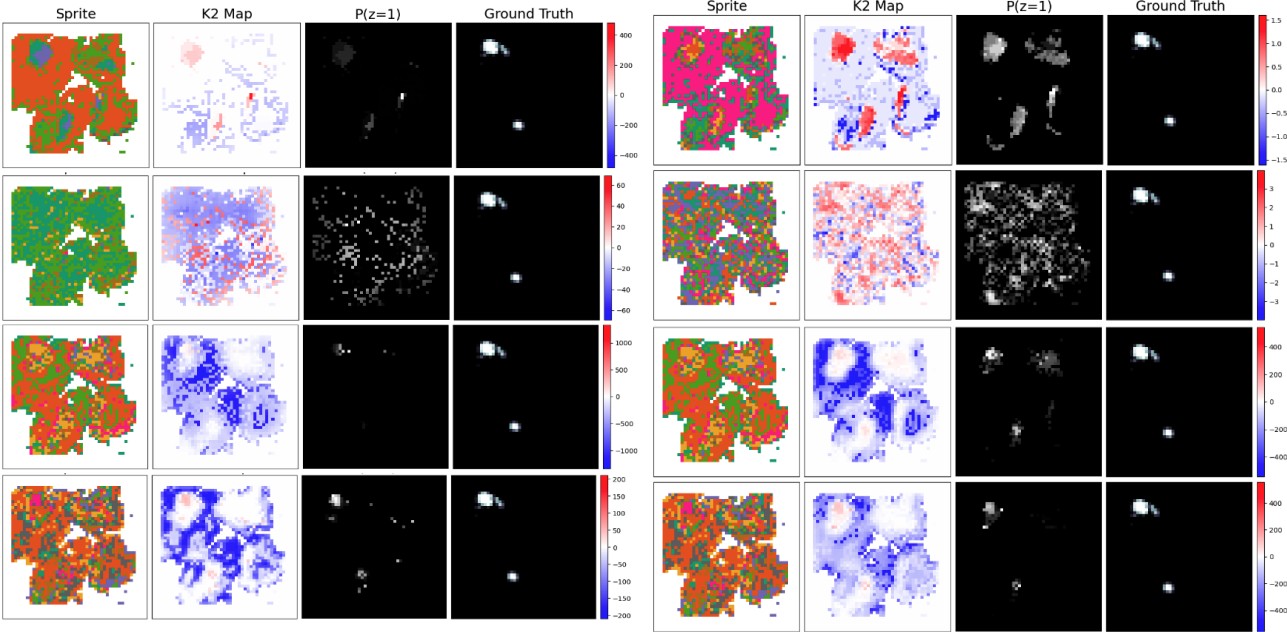

**Figure A.9:** Highest accuracy configurations.

**Figure A.10:** Highest Dice configurations.

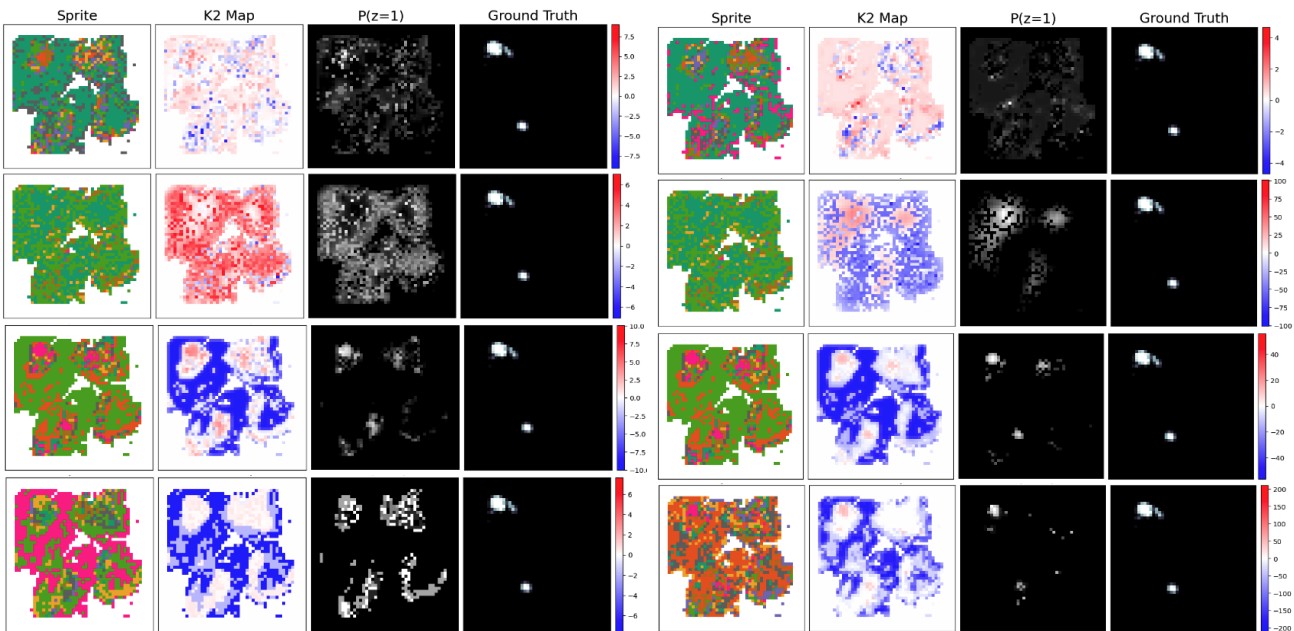

**Figure A.11:** Highest recall configurations.

**Figure A.12:** Highest balanced accuracy configurations.

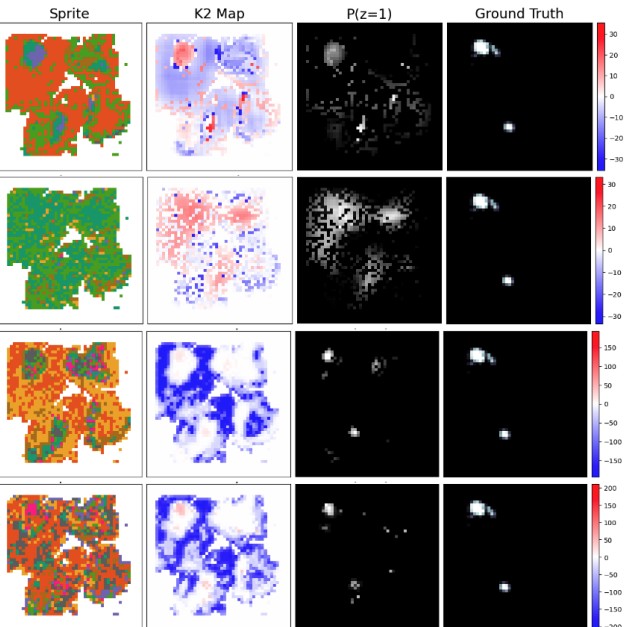

**Figure A.13:** Highest MCC configurations.

## A.7 Additional scatter plots for image classification vs SOD

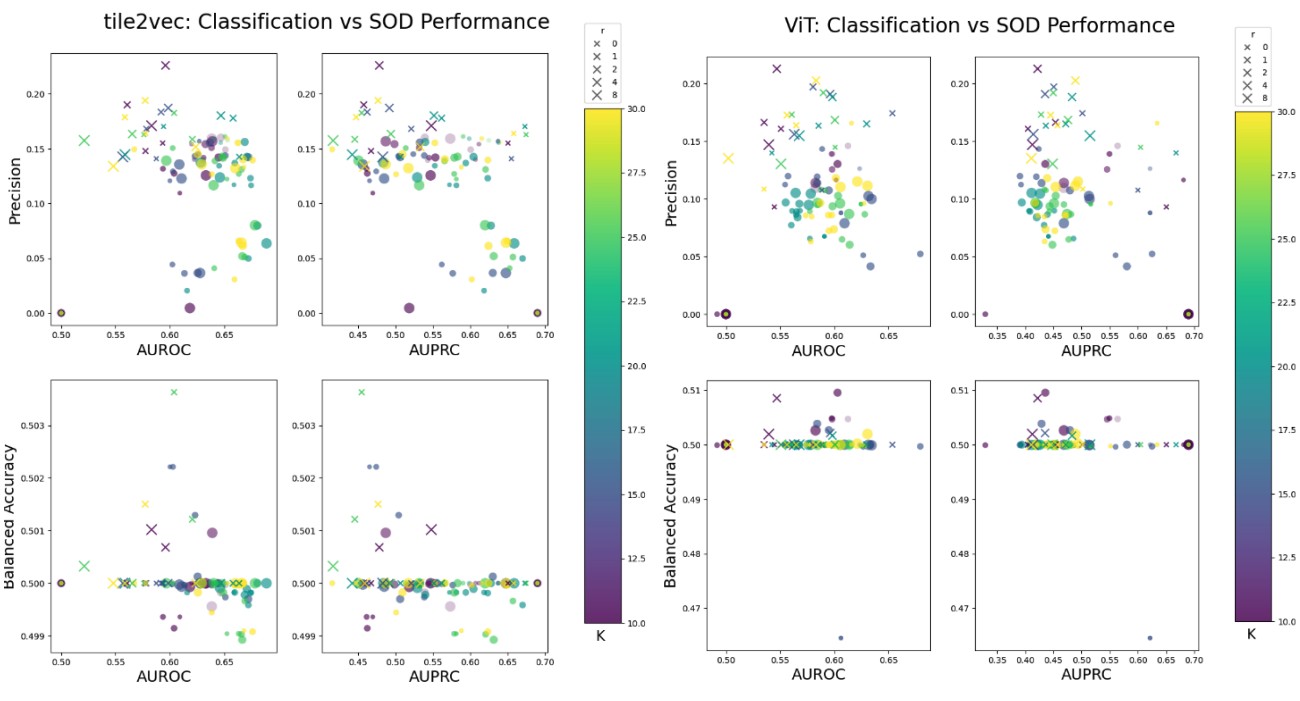

Figure A.14

Figure A.15

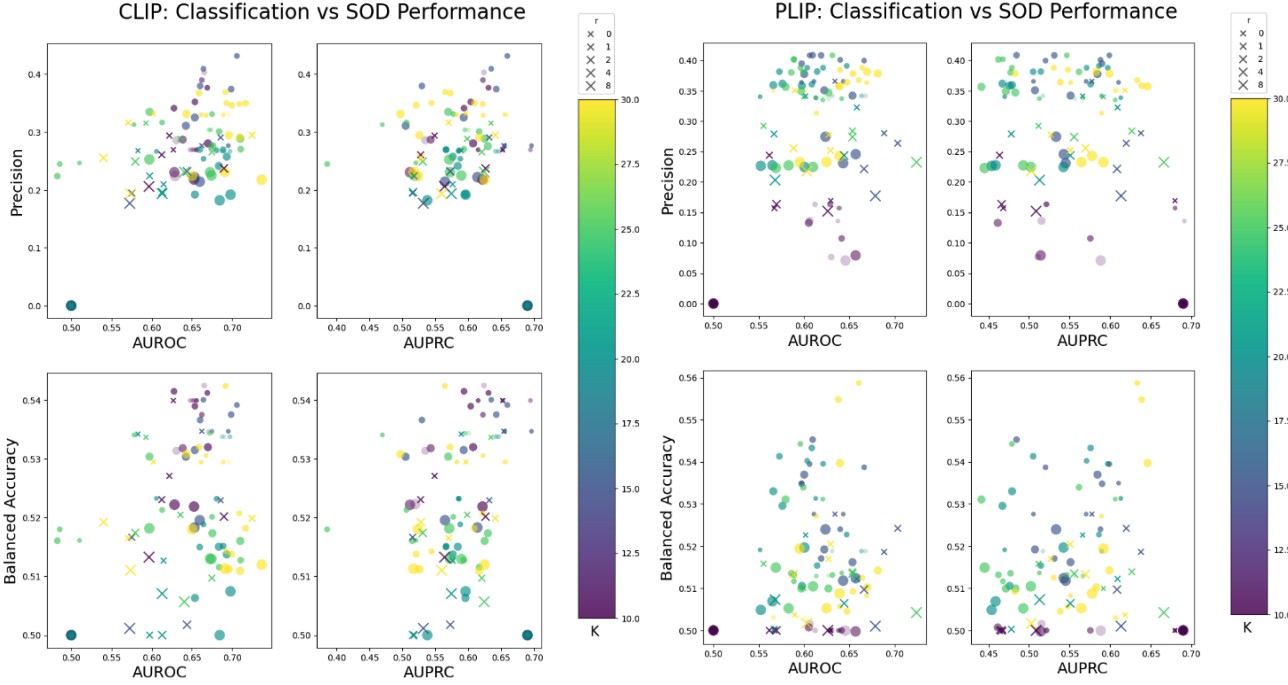

Figure A.16

Figure A.17

