# OpenReview forum: "Prospectors: Leveraging Short Contexts to Mine Salient Objects in High-dimensional Imagery"
_ICML.cc/2023/Workshop/IMLH — IMLH 2023 Poster_

### Official Review · Reviewer_S6n8 · 2023-06-12
**Good paper**

**Rating:** 8
**Confidence:** 4

**Review:**

This 8-page long paper proposes a formalized evaluation framework to assess visual grounding and presents prospector heads to improve visual grounding based on motif graph and bagged datum representation. The method is adaptable with various architectures and archives superior VG performances.

Strengths:

1) The paper is interesting and well-written. The introduction section is easy to follow.

2) The method is intuitive and effective. I recommend the authors continue diving into graph analysis for more insights.

Weakness:

1) The authors propose a complicated method but do not provide a thorough ablation analysis.

---

### Official Review · Reviewer_mQZ8 · 2023-06-14
**This is a well-developed paper, but there is still room for improvement in the performance and intention for method**

**Rating:** 6
**Confidence:** 3

**Review:**

This work first proposes an evaluation framework for high-dimensional visual grounding tasks based on adaptive and multiple thresholding and then presents a prospector head for grounding in the high-dimensional case.

***Strengths***:

Quality: The overall quality of this work is satisfactory. The length of the paper meets the requirements for a full conference paper. The method and experiments are solid.

Clarity: The presentation of the motivation and approach to the article is clear, and the paper is easy to read and easy to catch the point the author is trying to make.

Originality: The authors focus on a novel question in weakly supervised learning, the salient object detection in real-world and high-dimensional settings. The proposed evaluation method and prospector architecture make sense in this new case and are also novel.

Significance: The problem to be solved in this paper has some significance in the field of weakly supervised learning, which is solving the problem of not being able to evaluate SOD on large images of real scenes, and the problem of isolation between image chunks when processing SOD.

***Weaknesses***:

Quality: The method, prospector, proposed in this work does not achieve significant performance gain in Table 2. In addition, the proposed method and evaluation metric are quite complex and need released codes for reproducibility.

Clarity: Prospector is the main contribution in this work, but the authors describe it after the new evaluation metric, making the logic a little strange from the reader's point of view. Moreover, it is recommended to summarize the main challenge and the solution in a single figure as motivation instead of Figures 1 and 2 in the paper. Finally, the reason for naming the prospector as “K2” is puzzling.

Originality: The proposed evaluation metric is similar to Machiraju, G., et al. 2022. The prospector head is quite complex, but the authors didn’t mention any of the references or motivations of its key innovations (e.g., graphical data structure).

Significance: I am not sure if evaluating the SOD for large images has any practical meaning, and whether the previous approach of not using a high-dimensional setting was a big misstep. The methods evaluated in this work in Table 2 seem to be unstable, prospector seems do not have a significant improvement statistically. Furthermore, the authors should emphasize the significance of the paper in the field of interpretable ML for healthcare.

---

### Official Review · Reviewer_cyY5 · 2023-06-15
**Needs better presentation and explanations for the experiments**

**Rating:** 5
**Confidence:** 3

**Review:**

The author presents prospector heads for visual grounding. The proposed leverages chunk heterogeneity to identify salient objects over long ranges and can work under any image encoder.  Overall the paper is not well-written and misses several key pieces of information to understand its methodology. Please see below for detailed comments.

* How prospectors handle heterogeneity of the chunk? This is one of the key innovations highlighted by the author. However, the author only mentioned, "For intuition, heterogeneity of the sprite is parameterized by the choice of k in Step I." From this, it is hard to understand how it actually works and why it is heterogenous and can work better compared with homogeneous methods. More discussions and explanations are needed to understand it.

* The methodology Section 3.3 is not well-written. Several key components for understanding the methods are not well-presented. For example, What is a king's graph in line 242?

* Experiment results in Table 2 are confusing. Compared with SPM, k2 show poorer performance for several encoders. If the proposed method can only show effectiveness for certain metrics. More explanations and discussions are needed.Otherwise, it is hard to evaluate its effectiveness.

---

### Official Review · Reviewer_oBAQ · 2023-06-18
**Interesting work to improve salient object detection**

**Rating:** 6
**Confidence:** 3

**Review:**

This work proposes a formalized evaluation framework to assess visual grounding in high-dimensional image applications and presents prospector heads, a novel class of adaptation architectures designed to improve visual grounding. Results show that prospectors can enable many classes of encoders to identify salient objects without re-training and also demonstrate their improved performance against classical explanation techniques. The method is well-motivated and the results are good.

---

### Meta-Review · Area_Chair_5zgK · 2023-06-20

**Recommendation:** Accept (Poster)
**Confidence:** 5

**Metareview:**

Three of the four reviewers expressed positive opinions about this paper, appreciating its overall clarity and extensive experimentation. However, major weaknesses were detected, including the clarify of method, missing ablation study, and the originality of evaluation metrics. I kindly ask the authors to carefully consider the identified shortcomings and ensure that these issues are addressed in the final version.

---

### Decision · Program_Chairs · 2023-06-20

Accept (Poster)